# Partial Parameter Updates for Efficient Distributed Training

## Abstract

We introduce a memory- and compute-efficient method for low-communication distributed training. Existing methods reduce communication by performing multiple local updates between infrequent global synchronizations. We demonstrate that their efficiency can be significantly improved by restricting backpropagation: instead of updating all the parameters, each node updates only a fixed subset while keeping the remainder frozen during local steps. This constraint substantially reduces peak memory usage and training FLOPs, while a full forward pass over all parameters eliminates the need for cross-node activation exchange. Experiments on a 1.3B-parameter language model trained across 32 nodes show that our method matches the perplexity of prior low-communication approaches under identical token and bandwidth budgets while reducing training FLOPs by 15% and peak memory by up to 47%.

## 1 Introduction

Recent research has consistently shown that scaling language models (LLMs) improves their generalization and downstream capabilities (Yang et al., 2025; Team et al., 2025; Liu et al., 2024; Grattafiori et al., 2024).

At scale, training is typically achieved by distributing data across many compute nodes and synchronizing gradients at every optimization step. This synchronization relies on high-bandwidth interconnects, limiting large-scale training to high-end clusters with large number of well-connected nodes, a resource still accessible to only a small fraction of the machine learning community.

To reduce this dependence on high-bandwidth interconnects, prior work has explored three main directions. The first reduces the amount of data exchanged between nodes, for example through gradient sparsification, compression, or quantization (Alistarh et al., 2017; Lin et al., 2018; Tang et al., 2021; Shi et al., 2019). The second aims to hide communication latency by overlapping it with computation (Cohen et al., 2021; Sun et al., 2024; Kale et al., 2025), often by using delayed gradients combined with correction terms to preserve convergence. The third line of research, which our paper builds upon, lowers communication overhead by reducing the frequency of gradient synchronization. This approach, first introduced in the federated learning setting (McMahan et al., 2017), allows each model replica to perform multiple local updates before a global parameter average. Subsequent works have proposed more sophisticated methods for global synchronization, such as treating aggregated local differences as a pseudo-gradient for outer optimizer (Wang et al., 2019; Sun et al., 2022).

More recently, DiLoCo (Douillard et al., 2023) applies this dual-optimization scheme to LLM training, reducing bandwidth requirements by orders of magnitude compared to standard every-step gradient reduction. Streaming DiLoCo (Douillard et al., 2025) extends this idea by synchronizing only a subset of parameters at a time, thereby lowering both peak bandwidth and memory usage.

In low-bandwidth environments, memory-sharding approaches such as FSDP (Zhao et al., 2023) are impractical, since they require frequent communication across nodes that becomes prohibitively slow without fast interconnects. As a result, each device must store weights, gradients, and optimizer states locally, making memory the primary bottleneck (§ 3.1). These communication constraints also prevent the use of tensor parallelism (Shoeybi et al., 2019), which relies on synchronization at every step to reduce per-device computation.

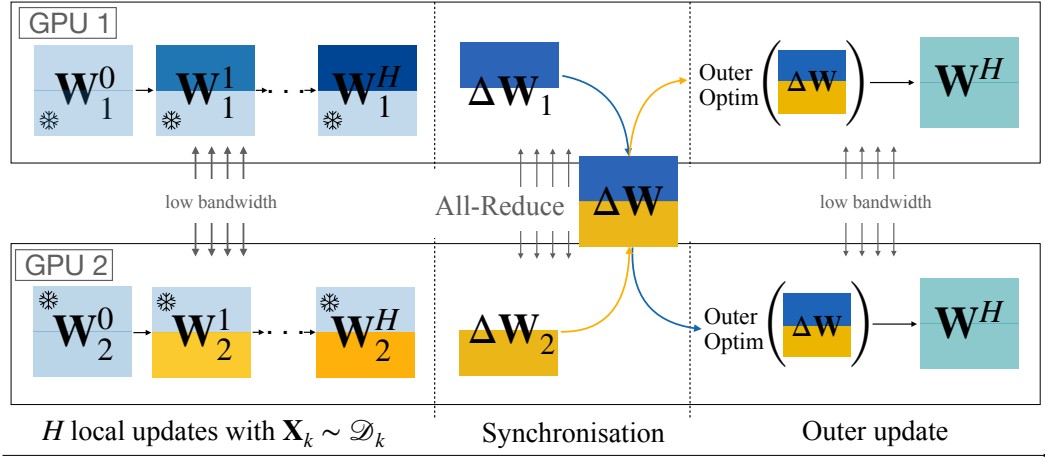

Figure 1: **Partial Parameter Updates.** Illustration of our low-communication distributed training procedure in a two-node setup connected by a low-bandwidth interconnect. Each node $k$ starts with an identical replica of the parameter matrix $\mathbf{W}_k$. During local training, each GPU updates only a disjoint slice of $\mathbf{W}_k$ while keeping the remaining parameters frozen. After $H$ local steps, parameter updates are synchronized via an all-reduce, and an outer optimizer step is applied to the previously frozen slices. This process repeats until convergence.

To address these limitations, we propose a simple yet effective alternative that improves both memory efficiency and training FLOPs without introducing frequent synchronization. Our approach can be viewed as distributed block coordinate optimization: each node backpropagates through and updates only a fixed slice of the parameters, treating the remainder as constant. After several local steps, parameter differences are averaged across nodes followed by an outer optimizer step (Figure 1). By restricting both backpropagation and optimizer updates to the active slice, our method reduces peak memory usage and total training FLOPs, while maintaining the low communication requirements and final performance of prior works.

Our main contributions are as follows:

- We introduce an efficient algorithm for low-communication distributed data-parallel training that performs local updates on a node-specific subset of parameters, thereby reducing both memory usage and computational cost (Algorithm 1).

- We empirically validate the effectiveness of our method by training a 1.3B-parameter language model on 32 nodes, achieving perplexity comparable to prior low-communication training approaches under the same token and bandwidth budgets, while using 15% fewer FLOPs and up to 47% less memory (Figure 2).

- We demonstrate that in simulated low-bandwidth settings, our method converges substantially faster than standard distributed data parallel training with every step synchronization (Figure 3).

## 2 METHOD

In this section, we formalize our proposed method for low-communication training. We begin in § 2.1 with a brief overview of language modeling and distributed data parallelism in both high- and low-bandwidth settings. In § 2.2, we then present the core idea of our method, followed by its training procedure and implementation details.

## 2.1 BACKGROUND

**Language Modeling**   Let $\mathcal{D}$ be a dataset of token sequences $\mathbf{x} = (x_1, \ldots, x_S)$ with $x_s \in \mathcal{V}$, where $\mathcal{V}$ is the vocabulary and $S$ is the sequence length. Language modeling aims to learn the data distribution $p(\mathbf{x})$, which can be factorized autoregressively as: $p(\mathbf{x}) = \prod_{s=1}^{S-1} p(x_{s+1} \mid \mathbf{x}_{1:s}; \theta)$ where $\theta$ denotes the model parameters. The parameters are typically estimated by minimizing the expected negative log-likelihood over the dataset:

$$\theta^\star = \arg\min_\theta \ \mathbb{E}_{\mathbf{x} \sim \mathcal{D}} \ \mathcal{L}(\mathbf{x}; \theta), \tag{1}$$

$$\mathcal{L}(\mathbf{x}; \theta) = -\sum_{s=1}^{S-1} \log p(x_{s+1} \mid \mathbf{x}_{1:s}; \theta). \tag{2}$$

In practice, this objective is minimized using a variant of stochastic gradient descent, where at each step the gradient $\nabla_\theta \mathcal{L}(\mathbf{X}; \theta)$ is computed on a mini-batch of sequences $\mathbf{X}$.

**Distributed Data Parallelism (DDP)**   To scale the optimization in Eq. 1, a common approach is to partition dataset $\mathcal{D}$ across $K$ compute nodes, with each node $k$ holding a shard $\mathcal{D}_k$. At each training step $t$, every node $k$ computes a gradient on its local mini-batch $\mathbf{X}_k^{(t)} \sim \mathcal{D}_k$:

$$g_k^{(t)} \ = \ \nabla_\theta \mathcal{L}(\mathbf{X}_k^{(t)}; \theta^{(t)}).$$

These local gradients are aggregated via an `All-Reduce` collective operation (Patarasuk & Yuan, 2009) to form $g^{(t)} \ = \ \frac{1}{K} \sum_{k=1}^{K} g_k^{(t)}$, which is then used to update the model parameters on all nodes. The model parameters, optimizer states, and gradients may be fully replicated on each node or partitioned to reduce memory usage (Zhao et al., 2023; Rajbhandari et al., 2020).

**Low-communication Distributed Data Parallelism**   Standard DDP communicates gradients at every step, making it impractical on hardware lacking high-bandwidth, low-latency interconnects. Low-communication methods relax this requirement by reducing the synchronization frequency.

A training round (global step) $t$ begins with all $K$ nodes holding identical global parameters $\theta^{(t)}$. Each node $k$ then performs $H$ local updates independently using an inner optimizer. At each local step $h = 0, \ldots, H-1$, node $k$ computes a gradient $g_k^{(t,h)}$ on its local mini-batch and applies the inner update:

$$\theta_k^{(t,h+1)} \ \leftarrow \ \text{INNEROPT}\big(\theta_k^{(t,h)}, \ g_k^{(t,h)}\big). \tag{3}$$

After $H$ local steps, each node computes its parameter delta relative to the starting point and participates in an all-reduce to compute the average update:

$$\Delta^{(t)} \ = \ \frac{1}{K} \sum_{k=1}^{K} \big(\theta_k^{(t,H)} - \theta^{(t)}\big). \tag{4}$$

The global parameters are then updated via an *outer optimizer*:

$$\theta^{(t+1)} \ \leftarrow \ \text{OUTEROPT}\big(\theta^{(t)}, \ \Delta^{(t)}\big). \tag{5}$$

In practice, the outer optimizer may simply apply $\Delta\theta^{(t)}$ directly (McMahan et al., 2017) or interpret it as a pseudo-gradient for an optimizer such as SGD (Wang et al., 2019; Sun et al., 2022). For large-scale language model training, DiLoCo (Douillard et al., 2023) reports that using AdamW as the inner optimizer and Nesterov SGD (Nesterov, 2013) as the outer optimizer yields lower validation loss than other combinations.

**Memory Usage and Computational Costs**   Techniques designed to lower peak memory usage by sharding the optimizer state, gradients, and parameters across devices are impractical on hardware without high-speed interconnects, as they require all-gather and reduce-scatter at every optimization step (Zhao et al., 2023; Ren et al., 2021). In addition to model weights and gradients, the state of the outer optimizer (e.g., momentum) must also remain in device memory. While synchronizing only a subset of parameters at a time and offloading the remainder to the host can reduce peak

**Algorithm 1**

1: **Inputs:** outer rounds $T$, local steps $H$, number of nodes $K$
2: **Notation:** $I_k^{\text{train}} \subseteq \{1, \ldots, |\theta|\}$; $I_k^{\text{frozen}} = \{1, \ldots, |\theta|\} \setminus I_k^{\text{train}}$;
3: $\quad\quad$ count vector $\mathbf{m} \in \{0, \ldots, K\}^{|\theta|}$ with $\mathbf{m}[i] := \sum_{k=1}^{K} \mathbb{1}\{i \in I_k^{\text{train}}\}$
4: **for** $t = 0 \ldots T - 1$ **do**
5: $\quad$ **for** $k = 0 \ldots K - 1$ **do** $\quad\quad\quad\quad\quad\quad\quad\quad$ # Execute in parallel on $K$ nodes
6: $\quad\quad$ $\theta_k^{(t,0)} \leftarrow \theta^{(t)}$
7: $\quad\quad$ **for** $h = 0 \ldots H - 1$ **do** $\quad\quad\quad\quad$ # Perform $H$ local steps independently on each node
8: $\quad\quad\quad$ $\mathbf{X}_k^{(t,h)} \sim \mathcal{D}_k$
9: $\quad\quad\quad$ $g_k^{(t,h)}[i] = \begin{cases} \nabla_{\theta[i]} \mathcal{L}\left(\theta_k^{(t,h)}; \mathbf{X}_k^{(t,h)}\right), & \text{if } i \in I_k^{\text{train}} \\ 0, & \text{otherwise} \end{cases}$
10: $\quad\quad\quad$ $\theta_k^{(t,h+1)}[I_k^{\text{train}}] \leftarrow \text{INNEROPT}\left(\theta_k^{(t,h)}[I_k^{\text{train}}], g_k^{(t,h)}[I_k^{\text{train}}]\right)$
11: $\quad\quad$ **end for**
12: $\quad\quad$ $\Delta_k^{(t)}[i] = \begin{cases} \theta_k^{(t,H)}[i] - \theta^{(t)}[i], & i \in I_k^{\text{train}} \\ 0, & \text{otherwise} \end{cases}$
13: $\quad$ **end for**
14: $\quad$ $\Delta^{(t)}[i] = \frac{1}{\mathbf{m}[i]} \sum_{k=1}^{K} \Delta_k^{(t)}[i], \quad i = 1, \ldots, |\theta|$ $\quad$ # Element-wise average by count vector $\mathbf{m}$
15: $\quad$ $\theta^{(t+1)} \leftarrow \text{OUTEROPT}(\theta^{(t)}, \Delta^{(t)})$
16: **end for**

usage (Douillard et al., 2025), the overall footprint remains large. As a result, even a relatively modest 1.3B-parameter model with full activation checkpointing consumes roughly 18 GB of GPU memory when trained without sharding using Adam optimizer (Loshchilov & Hutter, 2017) (Fig. 2a, § 3.1).

Our objective is to reduce memory footprint and per node FLOPs without degrading model quality or increasing communication compared to existing low-communication methods. In practice, this enables billion-parameter training on commodity GPUs with limited memory, connected over Wi-Fi or Ethernet.

## 2.2 PARTIAL PARAMETER UPDATES

Our method can be viewed as a distributed variant of block coordinate descent: on each node $k$, we partition the model parameters $\theta$ into a **trainable parameters**, indexed by a fixed set $I_k^{\text{train}} \subseteq \{1, \ldots, |\theta|\}$, and a **frozen parameters**, indexed by its complement $I_k^{\text{frozen}}$. As discussed in § 2.2.1, the trainable parameter sets assigned to different nodes overlap, i.e., $I_i \cap I_j \neq \emptyset$, for some $i, j \in \{0, \ldots, K-1\}$.

During local training, node $k$ only computes gradients for and applies updates to its designated trainable slice. The training process for a local step $h$ (within a global step $t$) on node $k$ proceeds as follows. The forward pass is standard, using the full local parameters $\theta_k^{(t,h)}$. The backward pass, however, is modified to compute gradients only for the trainable parameters (line 9):

$$g_k^{(t,h)}[i] = \begin{cases} \nabla_{\theta[i]} \mathcal{L}(\theta_k^{(t,h)}; \mathbf{X}_k^{(t,h)}), & \text{if } i \in I_k^{\text{train}} \\ 0, & \text{otherwise.} \end{cases}$$

The inner optimizer then updates only the active parameters corresponding to these non-zero gradients (line 10):

$$\theta_k^{(t,h+1)}[I_k^{\text{train}}] \leftarrow \text{INNEROPT}(\theta_k^{(t,h)}[I_k^{\text{train}}], g_k^{(t,h)}).$$

After $H$ local steps, the nodes synchronize. First, each node $k$ computes its local parameter delta, which is also non-zero only on its trainable slice (line 12):

$$\Delta_k^{(t)}[i] = \begin{cases} \theta_k^{(t,H)}[i] - \theta^{(t)}[i], & \text{if } i \in I_k^{\text{train}} \\ 0, & \text{otherwise.} \end{cases}$$

Next, these sparse deltas are aggregated across all nodes using an `All-Reduce` operation to form: $\Delta^{(t)} = \sum_{k=1}^{K} \Delta_k^{(t)}$. Finally, this summed delta is normalized element-wise by a count vector $\mathbf{m} \in \{1, \ldots, K\}^{|\theta|}$, where $\mathbf{m}[i]$ is the number of nodes responsible for updating parameter $\theta[i]$ (line 14). The normalized update is then applied by the outer optimizer (line 15). The full training procedure is detailed in Algorithm 1.

This design offers two benefits: (i) reduced per-node memory usage, as no gradient buffers or optimizer state are allocated for parameters in $I_k^{\text{frozen}}$ (Figure 2a), and (ii) lower training FLOPs, since gradients for $\theta[i]$ with $i \in I_k^{\text{frozen}}$ are never computed (Figure 2b, Appendix C). In § 3.2, we demonstrate that despite fewer updates per parameter than full-model baselines, our method achieves comparable perplexity.

### 2.2.1 PARAMETER SLICING

The assignment of trainable parameters $I_k^{\text{train}}$ to each node is controlled by a hyperparameter $N$, which specifies the number of distinct parameter slices. We assume that the total number of nodes $K$ is a multiple of $N$. Each node $k$ is assigned a slice index $n = k \bmod N$. This assignment determines how many nodes participate in updating each parameter block, which is captured by the count vector $\mathbf{m}$ (line 12):

$$\mathbf{m}[i] = \begin{cases} \frac{K}{N}, & i \in I^{\text{train}} \\ K, & \text{otherwise} \end{cases}.$$

We consider two strategies for partitioning the parameters into trainable and frozen subsets.

**MLP-Only Slicing** We slice only the MLP blocks, while all other parameters (attention, embeddings, normalization layers) are trained on all $K$ nodes. The rationale is that MLPs contain the majority of a Transformer's parameters, and when sliced, each block can be treated as an independent feed-forward pathway (similar in spirit to a Mixture-of-Experts layer (Shazeer et al., 2017)). This makes the partitioning straightforward both conceptually and in implementation.

An MLP block is typically defined as: $\text{MLP}(\mathbf{x}) = \mathbf{V}(\text{ReLU}(\mathbf{W}\mathbf{x}))$, where $\mathbf{W} \in \mathbb{R}^{4d \times d}$ and $\mathbf{V} \in \mathbb{R}^{d \times 4d}$ are the up- and down-projection matrices, respectively. We partition $\mathbf{W}$ row-wise into $N$ blocks $\{\mathbf{W}_1, \ldots, \mathbf{W}_N\}$ and $\mathbf{V}$ column-wise into $\{\mathbf{V}_1, \ldots, \mathbf{V}_N\}$, where $\mathbf{W}_n \in \mathbb{R}^{(4d/N) \times d}$ and $\mathbf{V}_n \in \mathbb{R}^{d \times (4d/N)}$. The MLP computation can then be expressed as a sum over these slices:

$$\text{MLP}(\mathbf{x}) = \sum_{n=1}^{N} \mathbf{V}_n(\text{ReLU}(\mathbf{W}_n \mathbf{x})).$$

On a given node $k$ with slice index $n$, the trainable parameters $I_k^{\text{train}}$ consist of all non-MLP parameters plus the specific MLP slices $\{\mathbf{W}_n, \mathbf{V}_n\}$ from every layer. The remaining $N-1$ MLP slices are kept frozen.

**Slicing MLPs and Attention Heads** We further extend the MLP-only slicing strategy by applying partial updates to the multi-head attention (MHA) block. In a standard MHA block (Vaswani, 2017), the input is projected by the query, key, and value matrices: $\mathbf{W}_Q, \mathbf{W}_K, \mathbf{W}_V \in \mathbb{R}^{d \times (h \cdot d_h)}$, where $h$ is the number of heads and $d_h$ the per-head dimension (so that $d = h \cdot d_h$). Then the concatenated head outputs are projected by a final matrix $\mathbf{W}_O \in \mathbb{R}^{(h \cdot d_h) \times d}$. We slice the input projections only since extending it to the entire attention block (including the output projection) led to noticeable performance degradation (Appendix C).

We divide the $h$ total attention heads into $N$ disjoint groups of size $h/N$. For node $k$, the assigned slice index is $n = k \bmod N$, with head group:

$$\mathcal{H}_n = \{n \cdot (h/N), \ldots, (n+1) \cdot (h/N) - 1\}.$$

On this node, the trainable attention parameters are limited to the columns:

$$\mathbf{W}_Q^{(n)} = \mathbf{W}_Q[:, \mathcal{H}_n], \mathbf{W}_K^{(n)} = \mathbf{W}_K[:, \mathcal{H}_n], \mathbf{W}_V^{(n)} = \mathbf{W}_V[:, \mathcal{H}_n].$$

All other columns in these three projections are kept frozen.

## 3 EXPERIMENTS

In this section, we present an empirical evaluation of our method. In § 3.1 we describe the experimental setup and explain how we measure memory usage and communication overhead. In § 3.2 we first compare our method to Streaming DiLoCo (Douillard et al., 2025) in terms of memory consumption and total training FLOPs, showing that our approach achieves comparable test perplexity while using fewer FLOPs, less memory, and the same bandwidth. We then demonstrate that low-communication methods (including ours), although requiring more training tokens to reach a target test loss, achieve shorter wall-clock training time than standard Distributed Data Parallel (DDP) with full gradient synchronization. Finally, we analyze how varying the size of the parameter subset updated on each node affects both test perplexity and memory usage.

### 3.1 EXPERIMENTAL SETUP

We use the RedPajama-V2 dataset (Weber et al., 2024), which consists of data from different sources, including Arxiv, Common Crawl, GitHub, and Wikipedia. In all experiments we use sequences of 1,024 tokens. n our experiments, we use Transformer models (Vaswani, 2017) with 1.3B and 2.6B parameters, matching the GPT-3 architectures (Brown et al., 2020): the 1.3B model has 24 layers with a hidden size of 2048, and the 2.6B model has 32 layers with a hidden size of 2560. We use rotary positional encodings (Su et al., 2024) and a SentencePiece tokenizer (Kudo & Richardson, 2018) with a vocabulary size of 32,000.

All models are trained using the AdamW optimizer (Loshchilov & Hutter, 2017) with $\beta_1 = 0.9$, $\beta_2 = 0.99$, and a weight decay of $0.1$. The learning rate is linearly warmed up to $3 \times 10^{-4}$ over the first $1,500$ steps, followed by cosine decay.

For both our method and Streaming DiLoCo, we adopt the outer optimization setup of Douillard et al. (2025): SGD with Nesterov momentum ($m = 0.9$) (Nesterov, 2013), learning rate $4 \times 10^{-1}$, and synchronization frequency $H = 100$. We also follow their streaming synchronization scheme: the 24 layers are divided into 8 groups of 3 layers each synchronized every $H$ local steps (see Appendix A.1).

We train with a batch size of $512$, distributed across 32 NVIDIA H100 GPUs (80GB each) for 1.3B model and $64$ H100 for 2.6B, resulting in a per-GPU batch size of $16$. Each GPU is treated as an independent compute node; we do not assume faster communication within an 8-GPU server.

**Memory Usage** We assume training in `bfloat16` with full activation recomputation during the backward pass. In mixed-precision training, a master copy of model parameters is typically maintained in `float32` for updates, with parameters cast to `bfloat16` on the fly for forward and backward computation. Gradients are stored in `bfloat16`, while optimizer states remain in `float32`. For low-bandwidth training with an outer optimizer, additional memory must be reserved for offloaded weights and momentum buffers. When synchronization is performed in a streaming fashion (grouped communication, as in Douillard et al. (2025)), this additional overhead is relatively small (see Figure 2a).

With activation recomputation, peak memory is dominated by: (i) optimizer states, (ii) weights, (iii) gradients, (iv) outer-optimizer states (if used), and (v) offloaded parameters (if any), as illustrated in Figure 2a.

**Communication Overhead** Let $M$ denote the total gradient size (in bytes), $K$ the number of nodes, and $B$ the peak per-link bandwidth. We assume that gradient synchronization is performed using a bandwidth-optimal ring all-reduce, implemented as a reduce–scatter followed by an all–gather (Thakur et al., 2005). Under bandwidth-optimality assumption, each node transmits a total of $2\frac{K-1}{K}M$ bytes per synchronization, leading to the following estimate of communication time: $T_{\text{comm}} \approx \frac{2(K-1)}{K}\frac{M}{B}$.

This estimate is a lower bound since it assumes that each link achieves its peak bandwidth with perfect overlap of send and receive operations, and it neglects non communication overhead such as kernel launch latency, stream synchronizations.

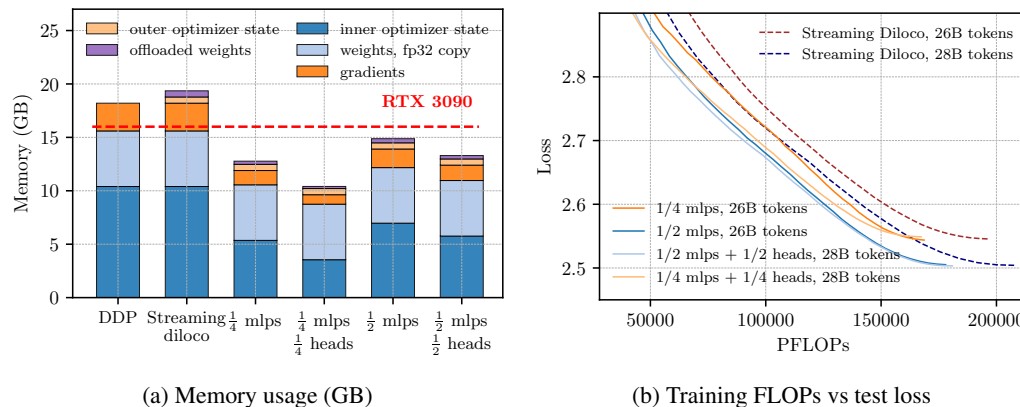

(a) Memory usage (GB)    (b) Training FLOPs vs test loss

Figure 2: **Less memory, fewer FLOPs, same performance.** Comparison of memory usage and total training FLOPs between our approach and Streaming DiLoCo. In each Transformer layer we either slice only the MLPs ($\frac{1}{N}$ MLPs) or slice both MLPs and attention heads ($\frac{1}{N}$ MLPs, $\frac{1}{N}$ heads). In both cases, only $1/N$ of the parameters in the corresponding projections are trained on each node (§ 2.2.1). **(a)** Estimated memory usage for DDP, Streaming DiLoCo, and our four variants (§ 3.1). **(b)** Test perplexity as a function of total training FLOPs for our method and to Streaming DiLoCo (§ 3.2, Appendix C)

## 3.2 RESULTS

**Peak Memory Footprint** Figure 2a demonstrates that our method requires significantly less memory than Streaming DiLoCo and DDP. This reduction comes from the fact that we do not train a large portion of parameters (detailed in Table 3.2), which means we neither maintain optimizer state nor store gradients for these parameters. For instance, $1/4$ mlps + $1/4$ heads configuration of our method uses $47\%$ less memory compared to full model training, while achieving similar test loss. This allows us to fit training with activation checkpointing of a 1.3B model using devices with less than 16GB of RAM.

**Compute Efficiency** We compare our method to Streaming DiLoCo in terms of training FLOPs. Figure 2b shows test loss as a function of total training FLOPs. For this comparison, we trained the $1/4$-MLP, $1/2$-MLP, and Streaming DiLoCo configurations with the Chinchilla-optimal token budget (26B). To match the performance of the $1/2$-MLP configuration, we slightly increased the token budget for Streaming DiLoCo to 28B. We also trained the $1/N$-MLP+$1/N$-heads configurations on 28B tokens to match the performance of their corresponding $1/N$-MLP runs. Across these performance-matched comparisons, our method consistently required $15\%$ fewer total FLOPs.

**Convergence Speed Under Bandwidth Constraints** We compare our method with Streaming DiLoCo and standard DDP by simulating total training time under bandwidth-constrained conditions (Figure 3). While low-communication methods require more training tokens to achieve the same performance as Distributed Data Parallel (DDP), they are significantly faster in terms of wall-clock time on slow networks.

Our simulation model deliberately favors DDP by assuming perfect overlap between computation and communication, giving a per-step runtime of $T = \max(T_{\text{comm}}, T_{\text{comp}})$. In contrast, for low-communication methods we assume no overlap: $T = T_{\text{comm}} + T_{\text{comp}}$.

We intentionally model a best-case scenario for DDP to demonstrate that even when its step time is minimized, low-communication methods converge faster under bandwidth constraints settings (see Appendix B for details). Our method as Streaming DiLoCo uses identical bandwwidth budget (more details in Appendix B).

**Parameter Slicing** Table 3.2 reports the relation between the number of slices, memory usage, number of trainable parameters, and final loss. As expected, reducing memory by freezing more parameters per node leads to drops in performance. To better understand this trade-off, we overtrained

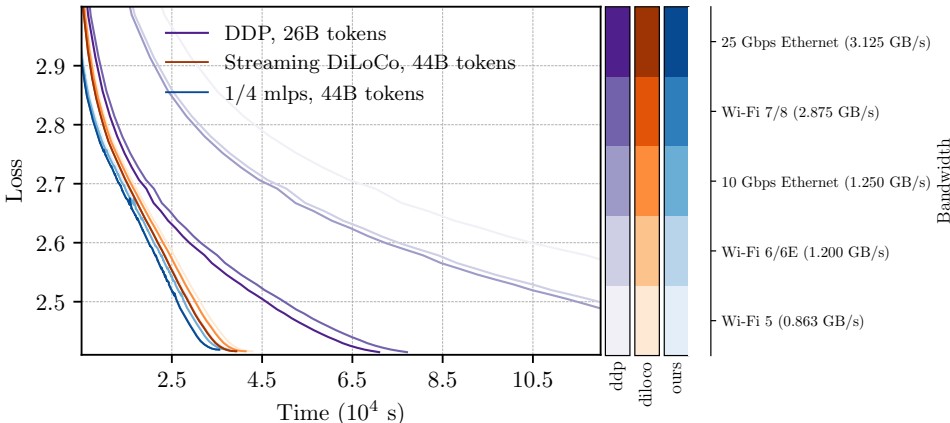

Figure 3: **Faster convergence without fast interconnects.** Simulated training time for our method, Streaming DiLoCo, and standard DDP under varying bandwidth limits. Blue, orange, and purple denote our method, Streaming DiLoCo, and DDP, respectively; transparency levels indicate different peak bandwidths (in GB/s). For DDP, step time is estimated as the maximum of single-GPU compute and gradient communication (perfect overlap), whereas for low-communication methods it is the sum (no overlap) (§ 3.2). Although low-communication methods require $1.7\times$ more tokens to reach a validation loss of 2.41, they complete training in significantly less wall-clock time when network bandwidth is limited.

| Method | Perplexity | Memory, GB | Trainable parameters, B | Tokens, B |
|---|---|---|---|---|
| Streaming DiLoCo | 12.75 | 19.36 | 1.3 | 26 |
| 1/2 mlps | 12.24 | 14.87 | 0.87 | 26 |
| 1/4 mlps | 12.72 | 12.77 | 0.67 | 26 |
| 1/8 mlps | 13.59 | 11.72 | 0.57 | 26 |
| 1/16 mlps | 14.21 | 11.19 | 0.52 | 26 |
| 1/8 mlps, overtrained | 12.68 | 11.72 | 0.57 | 37 |
| 1/2 mlps + 1/2 heads | 12.22 | 13.29 | 0.72 | 28 |
| 1/4 mlps + 1/4 heads | 12.79 | 10.41 | 0.44 | 28 |
| 1/4 heads | 12.83 | 16.95 | 1.07 | 26 |
| Streaming DiLoCo | 10.47 | 38.7 | 2.59 | 52 |
| 1/4 mlps | 10.49 | 25.55 | 1.34 | 52 |

Table 1: Comparison of perplexity, memory usage, number of trainable parameters, and training tokens across different methods for 1.3B and 2.6B parameter models trained on 32 and 64 GPUs on RedPajama-V2 dataset. Different background colors correspond to different model sizes.

the configuration with 8 slices (each slice is updated on 4 nodes out of 32) and found that it required almost 50% more tokens to match the performance of smaller-slice configurations. As shown in Table 3.2, freezing only MLPs is less effective for memory savings than freezing a combination of MLPs and attention heads. Moreover, when freezing only 1/4 of attention heads, the performance is similar to the 1/4 MLP slicing configuration, but the number of active parameters, and therefore the memory usage, is higher.

In our setup, the set of trainable parameters is fixed throughout training. While dynamically reassigning parameters could, in principle, improve convergence, it would require either replicating the full optimizer state on every node or transferring optimizer state whenever a parameter's owner changes, both of which eliminate the memory and communication benefits we target. Exploring lightweight forms of adaptive parameter assignment during training that preserve these benefits remains an open

direction for future work. We also evaluated alternative parameter assignment strategies; details are provided in Appendix A.2.

## 4 RELATED WORK

We review prior work in two areas most relevant to our contributions: methods for low-communication distributed training and approaches for improving memory and computational efficiency during training. For the latter, we focus on memory-efficient optimizers and tensor parallelism, which are most directly related to our method.

**Low-Communication Training**    Communication overhead in distributed data-parallel training has been tackled in three main ways: reducing the volume of data exchanged between nodes with gradient compression or quantization (Dettmers, 2015; Alistarh et al., 2017; Lin et al., 2018; Li et al., 2023), hiding latency by overlapping communication with computation (Cohen et al., 2021; Sun et al., 2024; Kale et al., 2025), and lowering frequency of communication by performing multiple local updates between synchronizations (McMahan et al., 2017; Wang et al., 2019; Sun et al., 2022; Douillard et al., 2023; 2025). We show that the latter can be made substantially more memory- and compute-efficient by restricting backpropagation to partial parameter subsets. The three strategies are complementary, and compression or overlap techniques can be applied together with our method to further reduce communication costs. More recently, Beton et al. (2025) proposed sparse parameter synchronization, which reduces communication by synchronizing only a random fraction of parameters at each step. While this lowers divergence across nodes, all parameters are still updated on every device, meaning each node must store the full optimizer state and perform full backpropagation. In contrast, our method updates only a fixed subset of parameters per node, which directly reduces both memory and compute.

Another line of work studies pipeline parallelism in slow-network settings (Huang et al., 2019), which requires inter-stage communication of activations in every step. To mitigate this communication overhead, recent methods propose compressing or quantizing activations (Wang et al., 2022; Ryabinin et al., 2023; Yuan et al., 2022; Ramasinghe et al., 2025). Unlike these approaches, which still depend on activation exchange, our method operates purely in the data-parallel regime and targets an orthogonal axis of parallelization, and could in principle be combined with pipeline parallelism and activation compression in large-scale settings.

**Memory and Compute Efficiency**    A large fraction of GPU memory during training is occupied by optimizer states, particularly for adaptive methods such as Adam (Loshchilov & Hutter, 2017), which maintain first- and second-order moments for every parameter. The main savings of our approach come from the fact that each node only updates a subset of parameters. As a result, momentum states for the remaining parameters do not need to be stored locally, yielding substantial memory savings. This is especially important in low-communication settings, where sharding optimizer states across devices is impractical due to the communication overhead it introduces. Several methods aim to reduce optimizer state memory directly. One strategy is to quantize optimizer states to lower precision, for example 8-bit quantization (Dettmers et al., 2021; Li et al., 2023). Another is to apply low-rank projections to compress gradients and optimizer states (Zhao et al., 2024). Parameter grouping has also been explored: Zhang et al. (2024) maintain a single momentum vector per block of parameters, while Han et al. (2025) combine grouping with quantization. Such efficient optimizers are orthogonal to our method and could be combined with it for further savings.

Another line of work distributes compute and memory through tensor parallelism, where large matrix multiplications are partitioned across GPUs and results are gathered after each operation (Shoeybi et al., 2019). Our method is conceptually related, but applies slicing only in the backward pass: each device updates a portion of the parameter matrix while still executing the full forward computation. In contrast to tensor parallelism, our approach avoids frequent all-to-all communication and therefore does not depend on high-bandwidth interconnects.

## 5 CONCLUSION

We proposed an efficient method for low-communication distributed training. The core design of our approach is partial backpropagation: only a subset of parameters is updated on each node, reducing per-device memory and compute while maintaining convergence. We have shown that, despite some parameters receiving fewer gradient updates, our method matches the performance of prior low-communication approaches under identical bandwidth and token budgets. Future directions include exploring alternative parameter-partitioning strategies and investigating different backpropagation sparsity patterns (Appendix D).

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

CONTENTS

## A   ABLATIONS

### A.1   STREAMING SYNCHRONIZATION

One way to reduce peak memory usage in low-communication distributed training is to lower the memory consumed by the outer optimizer state and offloaded parameters. When parameters are synchronized in groups with multiple local steps in between, it is unnecessary to keep the full optimizer state in memory at every step. Instead, only the states and parameters of the currently active group need to be loaded. Douillard et al. (2025) explored this idea by grouping parameters at the granularity of transformer layers.

We experimented with alternative grouping strategies. In particular, rather than grouping by layers, we grouped by parameter slices. Under the slicing strategy described in § 2.2.1, at step $t$ we all-reduce gradients and update all MLP slices $\mathbf{W}_0^l$ and $\mathbf{V}_0^l$ for $l \in \{0, \ldots, L\}$. At step $t + \tau$, we update $\mathbf{W}_1^l$ and $\mathbf{V}_1^l$; at step $t + 2\tau$, $\mathbf{W}_2^l$ and $\mathbf{V}_2^l$; and so on, until all slices are synchronized. We found that this strategy degraded performance (Table 5b): while grouping by layers had little to no impact on final accuracy, grouping by slices did. A likely explanation is that only part of each weight matrix is updated by the outer optimizer, and these updates are much larger than the small local changes made

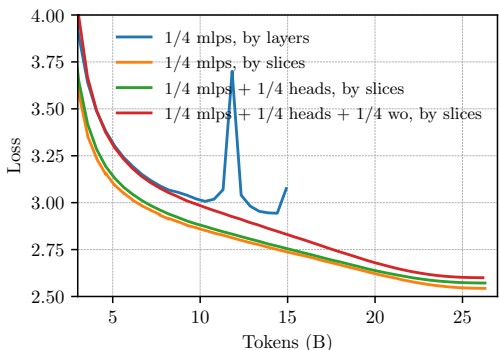

(a) Memory usage (GB)

| Method | Test perplexity |
|---|---|
| DiLoCo | 12.78 |
| Streaming DiLoCo | 12.75 |
| Ours ($\frac{1}{4}$ MLPs) | 12.73 |
| Ours (by slices, $\frac{1}{4}$ MLPs) | 13.47 |
| Ours (by layers, $\frac{1}{4}$ MLPs) | 12.72 |

(b) Test perplexity

Figure 5: **Streaming synchronization.** Comparison of memory usage and test perplexity with and without streaming synchronization for DiLoCo and our method on a 1.3B-parameter language model trained across 32 nodes. "By layers" means the 24 transformer layers are grouped sequentially into 8 groups of 3, plus a ninth group for embeddings and outer normalization. "By slices" means synchronization is performed by grouping MLP slices in each layer—4 groups, plus a 5th for embeddings and a 6th for attention and normalization layers. **(a)** Estimated memory usage per GPU (§ 3.1). **(b)** Final test perplexity after training on 26B tokens.

by the inner optimizer and probably such sudden change in only a part of matrix make the overall optimization problem more difficult.

In all our experiments, we adopt the streaming synchronization strategy of Douillard et al. (2025). Our method is orthogonal to this idea: our main contribution is reducing the memory footprint of the inner optimizer state and gradients. Streaming synchronization can be combined with our approach to further reduce memory usage (Figure 5a). Consistent with prior observations (Douillard et al., 2025), synchronization in groups does not affect final performance, either for our method or for DiLoCo (Table 5b).

Figure 4: Test loss as a function of training tokens for different variants of trainable parameter assignment. "By layers" corresponds to training only a subset of layers on each node – slicing model horizontally, whereas "by slices" refers to slicing parameters vertically as described in § 2.2.1.

### A.2 PARAMETER SLICING

In our main experiments, we considered two strategies for assigning trainable parameters to each node (§ 2.2.1): freezing parts of the MLPs, and freezing MLPs together with a part of attention heads. We also experimented with alternative slicing strategies. For instance, we attempted to slice the outer attention projection $\mathbf{W}_o$, but as shown in Figure 4, this led to some performance degradation.

Another variant we explored was training only a subset of layers on each node. Instead of slicing parameters vertically (by splitting weight matrices into slices), we partitioned the model horizontally, such that each node updates MLP layers parameters within a smaller set of layers. However, this proved to be a significantly harder optimization problem. We were unable to find a hyperparameter configuration that avoided gradient explosion, and training quickly diverged. An example training curve is shown in Figure 4. It is possible that with more extensive hyperparameter exploration or alternative stabilization techniques, this variant could be made to work, but we leave this for future investigation.

# B  COMMUNICATION OVERHEAD

We consider training a $1.3$B-parameter model in `bf16`, corresponding to $M = 2.6$ GB of gradients, on $K = 32$ nodes. Assume the nodes are connected via a high-speed Wi-Fi 7/8 network with a peak bandwidth of $B = 2.875$ GB/s. The per-step communication time can be approximated by

$$T_{\text{comm}} \approx \frac{2(K-1)}{K} \frac{M}{B}$$

Then for DDP:

$$T_{\text{comm}} \approx \frac{2(K-1)}{K} \frac{M}{B} \approx 1.75 \text{ s}.$$

This is nearly $4\times$ longer than the measured per-step compute time on a single H100 GPU ($\approx 0.44$ s), indicating that communication dominates overall step time even under optimistic peak-bandwidth assumptions.

In contrast, our method and Streaming DiLoCo synchronize only once every $S = 100$ steps, reducing the amortized communication cost to

$$T_{\text{low-comm}} = \frac{T_{\text{comm}}}{S} \approx \frac{1.75}{100} = 0.0175 \text{ s}.$$

As shown in Figure 3, this substantial reduction in communication time allows our method and Streaming DiLoCo to achieve roughly $2\times$ faster simulated wall-clock convergence than DDP at Wi-Fi 7 bandwidth, despite requiring more training tokens to reach comparable test loss (44B vs. 26B).

Our method requires the same bandwidth budget as Streaming DiLoCo. Although the outer gradients are sparse, ring all-reduce communicates full-sized tensors in multiple hops across devices, so the total amount of data exchanged, and thus the communication cost, remains unchanged.

# C  COMPUTATIONAL OVERHEAD

## C.1  PARTIAL BACKWARD

**MLP with frozen slices**   Consider a single MLP block with up-/down-projection matrices $\mathbf{W} \in \mathbb{R}^{4d \times d}$ and $\mathbf{V} \in \mathbb{R}^{d \times 4d}$, with an elementwise ReLU in between. Let the input activations for a batch/sequence be $\mathbf{X} \in \mathbb{R}^{d \times m}$ (feature dimension $d$, $m$ tokens).

We slice the hidden dimension into $N$ parts as described in § 2.2.1:

$$\mathbf{W} = \begin{bmatrix} \mathbf{W}_1 \\ \vdots \\ \mathbf{W}_N \end{bmatrix}, \quad \mathbf{W}_n \in \mathbb{R}^{\frac{4d}{N} \times d}, \qquad \mathbf{V} = \begin{bmatrix} \mathbf{V}_1 & \cdots & \mathbf{V}_N \end{bmatrix}, \quad \mathbf{V}_n \in \mathbb{R}^{d \times \frac{4d}{N}}.$$

Define the per-slice pre-activations and activations:

$$\mathbf{H}_n = \mathbf{W}_n \mathbf{X} \in \mathbb{R}^{\frac{4d}{N} \times m}, \qquad \mathbf{A}_n = \text{ReLU}(\mathbf{H}_n) \in \mathbb{R}^{\frac{4d}{N} \times m}.$$

Stacking along the hidden dimension gives $\mathbf{H} = \begin{bmatrix} \mathbf{H}_1 \\ \cdots \\ \mathbf{H}_N \end{bmatrix} \in \mathbb{R}^{4d \times m}$ and $\mathbf{A} = \begin{bmatrix} \mathbf{A}_1 \\ \cdots \\ \mathbf{A}_N \end{bmatrix} \in \mathbb{R}^{4d \times m}$.

The MLP forward then decomposes additively over slices:

$$\mathbf{Y} = \mathbf{V}\mathbf{A} = \sum_{n=1}^{N} \mathbf{V}_n \mathbf{A}_n \in \mathbb{R}^{d \times m}.$$

Given the upstream gradient $\mathbf{G} = \frac{\partial \mathcal{L}}{\partial \mathbf{Y}} \in \mathbb{R}^{d \times m}$, the backward pass:

1. **Output projection $\mathbf{V}_n$:**

$$\frac{\partial \mathcal{L}}{\partial \mathbf{V}_n} \;=\; \mathbf{G}\mathbf{A}_n^\top, \quad \mathbf{A}_n = \mathrm{ReLU}(\mathbf{W}_n\mathbf{X}).$$

2. **Activations $\mathbf{A}_n$:**

$$\frac{\partial \mathcal{L}}{\partial \mathbf{A}_n} \;=\; \mathbf{V}_n^\top\mathbf{G}.$$

3. **Through ReLU:**

$$\frac{\partial \mathcal{L}}{\partial \mathbf{H}_n} \;=\; \left(\tfrac{\partial \mathcal{L}}{\partial \mathbf{A}_n}\right) \odot \mathbb{I}(\mathbf{H}_n > 0), \quad \mathbf{H}_n = \mathbf{W}_n\mathbf{X}.$$

4. **Up-projection $\mathbf{W}_n$:**

$$\frac{\partial \mathcal{L}}{\partial \mathbf{W}_n} \;=\; \left(\tfrac{\partial \mathcal{L}}{\partial \mathbf{H}_n}\right)\mathbf{X}^\top.$$

5. **Input $\mathbf{X}$:**

$$\frac{\partial \mathcal{L}}{\partial \mathbf{X}} \;=\; \sum_{n=1}^{N} \mathbf{W}_n^\top \left(\tfrac{\partial \mathcal{L}}{\partial \mathbf{H}_n}\right).$$

As a result, since we do not update all the slices except $k$, we do not compute gradients with respect to the frozen weights. This yields FLOP savings, because we skip the multiplications needed to form

$$\frac{\partial \mathcal{L}}{\partial \mathbf{W}_n}, \quad \frac{\partial \mathcal{L}}{\partial \mathbf{V}_n}, \qquad \forall n \in \{1, \ldots, k-1, k+1, \ldots, N\}.$$

Note that we still need to compute the full Jacobian with respect to the input $\mathbf{X}$.

$$\frac{\partial \mathcal{L}}{\partial \mathbf{X}} = \sum_{n=1}^{N} \mathbf{W}_n^\top \left((\mathbf{V}_n^\top\mathbf{G}) \odot \mathbb{I}(\mathbf{H}_n > 0)\right),$$

where $\mathbf{H}_n = \mathbf{W}_n\mathbf{X}$ and $\mathbf{G} = \frac{\partial \mathcal{L}}{\partial \mathbf{Y}}$.

Even if the weights of slice $n$ are frozen, its contribution $\mathbf{W}_n^\top\left((\mathbf{V}_n^\top\mathbf{G}) \odot \mathbb{I}(\mathbf{H}_n > 0)\right)$ is still required to correctly propagate gradients to earlier layers (see Appendix D).

**MHA with Frozen Heads**  Recall the forward pass of multi-head attention:

$$\mathbf{Q} = \mathbf{W}_Q^\top\mathbf{X}, \quad \mathbf{K} = \mathbf{W}_K^\top\mathbf{X}, \quad \mathbf{V} = \mathbf{W}_V^\top\mathbf{X},$$

with $\mathbf{Q}, \mathbf{K}, \mathbf{V} \in \mathbb{R}^{hd_h \times m}$. We split them into $h$ heads:

$$\mathbf{Q} = \begin{bmatrix} \mathbf{Q}^{(1)} \\ \vdots \\ \mathbf{Q}^{(h)} \end{bmatrix}, \quad \mathbf{Q}^{(j)} \in \mathbb{R}^{d_h \times m},$$

and similarly for $\mathbf{K}^{(j)}$ and $\mathbf{V}^{(j)}$. Each head computes

$$\mathbf{S}^{(j)} = \tfrac{1}{\sqrt{d_h}}\mathbf{Q}^{(j)\top}\mathbf{K}^{(j)}, \quad \mathbf{A}^{(j)} = \mathrm{softmax}_{\mathrm{row}}(\mathbf{S}^{(j)}), \quad \mathbf{U}^{(j)} = \mathbf{V}^{(j)}\mathbf{A}^{(j)\top}.$$

Concatenate $\mathbf{U} = [\mathbf{U}^{(1)}; \ldots; \mathbf{U}^{(h)}] \in \mathbb{R}^{hd_h \times m}$, then project

$$\mathbf{Y} = \mathbf{W}_O^\top\mathbf{U} \in \mathbb{R}^{d \times m}.$$

Given upstream gradient $\mathbf{G} = \partial \mathcal{L}/\partial \mathbf{Y}$:

1. **Output projection.**

$$\frac{\partial \mathcal{L}}{\partial \mathbf{W}_O} = \mathbf{U}\mathbf{G}^\top, \quad \frac{\partial \mathcal{L}}{\partial \mathbf{U}} = \mathbf{W}_O\mathbf{G}.$$

2. **Per head $j$:**

$$\frac{\partial \mathcal{L}}{\partial \mathbf{V}^{(j)}} = \mathbf{G}_U^{(j)} \mathbf{A}^{(j)}, \quad \frac{\partial \mathcal{L}}{\partial \mathbf{A}^{(j)}} = \mathbf{G}_U^{(j)\top} \mathbf{V}^{(j)}.$$

3. **Through softmax:**

$$\frac{\partial \mathcal{L}}{\partial \mathbf{S}^{(j)}} = \text{softmax}\big(\mathbf{A}^{(j)}, \tfrac{\partial \mathcal{L}}{\partial \mathbf{A}^{(j)}}\big).$$

4. **Back to $\mathbf{Q}^{(j)}$ and $\mathbf{K}^{(j)}$:**

$$\frac{\partial \mathcal{L}}{\partial \mathbf{Q}^{(j)}} = \tfrac{1}{\sqrt{d_h}} \mathbf{K}^{(j)} \big(\tfrac{\partial \mathcal{L}}{\partial \mathbf{S}^{(j)}}\big)^\top, \quad \frac{\partial \mathcal{L}}{\partial \mathbf{K}^{(j)}} = \tfrac{1}{\sqrt{d_h}} \mathbf{Q}^{(j)} \big(\tfrac{\partial \mathcal{L}}{\partial \mathbf{S}^{(j)}}\big).$$

5. **Back to projection matrices.** Since $\mathbf{Q}^{(j)} = \mathbf{W}_Q^{(j)\top} \mathbf{X}$:

$$\frac{\partial \mathcal{L}}{\partial \mathbf{W}_Q^{(j)}} = \mathbf{X}\big(\tfrac{\partial \mathcal{L}}{\partial \mathbf{Q}^{(j)}}\big)^\top, \quad \frac{\partial \mathcal{L}}{\partial \mathbf{W}_K^{(j)}} = \mathbf{X}\big(\tfrac{\partial \mathcal{L}}{\partial \mathbf{K}^{(j)}}\big)^\top, \quad \frac{\partial \mathcal{L}}{\partial \mathbf{W}_V^{(j)}} = \mathbf{X}\big(\tfrac{\partial \mathcal{L}}{\partial \mathbf{V}^{(j)}}\big)^\top.$$

6. **Input gradient.**

$$\frac{\partial \mathcal{L}}{\partial \mathbf{X}} = \sum_{j=1}^{h} \Big( \mathbf{W}_Q^{(j)} \tfrac{\partial \mathcal{L}}{\partial \mathbf{Q}^{(j)}} + \mathbf{W}_K^{(j)} \tfrac{\partial \mathcal{L}}{\partial \mathbf{K}^{(j)}} + \mathbf{W}_V^{(j)} \tfrac{\partial \mathcal{L}}{\partial \mathbf{V}^{(j)}} \Big).$$

Let $\mathcal{H}_{\text{train}} \subseteq \{1, \dots, h\}$ be the set of trainable heads on this node (see § 2.2.1). Then for all $j \notin \mathcal{H}_{\text{train}}$,

$$\frac{\partial \mathcal{L}}{\partial \mathbf{W}_Q^{(j)}} = \frac{\partial \mathcal{L}}{\partial \mathbf{W}_K^{(j)}} = \frac{\partial \mathcal{L}}{\partial \mathbf{W}_V^{(j)}} = 0.$$

Gradients for $j \in \mathcal{H}_{\text{train}}$ are computed as above. Note that the input gradient $\partial \mathcal{L}/\partial \mathbf{X}$ still aggregates contributions from all heads, so freezing heads saves FLOPs only on parameter gradients computation.

## C.2   FLOPs CALCULATION

For a Transformer with batch size $B$, sequence length $S$, hidden dimension $H$, number of layers $L$, feedforward dimension $D_{\text{ff}}$, vocabulary size $V$.

### C.2.1   FORWARD

The forward FLOPs can be decomposed as:

- **Embedding Layer:** Although embeddings are typically implemented as lookups with negligible computational cost, for completeness, we estimate the FLOPs as:

$$\text{FLOPs}_{\text{emb}} = B \times S \times H$$

- **Multi-Head Attention (MHA):**

  1. Linear Projections (Queries, Keys, Values):

  $$\text{FLOPs}_{\text{proj}} = 3 \times 2 \times B \times S \times H \times H = 6 \times B \times S \times H^2$$

  2. Scaled Dot-Product Attention:

  $$\text{FLOPs}_{\text{attn}} = \text{FLOPs}_{\text{QK}} + \text{FLOPs}_{\text{V}} = 2 \times B \times S^2 \times H + 2 \times B \times S^2 \times H = 4 \times B \times S^2 \times H$$

  3. Output Projection:

  $$\text{FLOPs}_{\text{out\_proj}} = 2 \times B \times S \times H \times H$$

  Total Multi-Head Attention (MHA):

  $$\begin{aligned} \text{FLOPs}_{\text{MHA}} &= \text{FLOPs}_{\text{proj}} + \text{FLOPs}_{\text{attn}} + \text{FLOPs}_{\text{out\_proj}} \\ &= 6 \times B \times S \times H^2 + 4 \times B \times S^2 \times H + 2 \times B \times S \times H^2 \\ &= 8 \times B \times S \times H^2 + 4 \times B \times S^2 \times H \end{aligned}$$

- **Feedforward Network (FFN):**

$$\text{FLOPs}_{\text{FFN}} = 2 \times 2 \times B \times S \times H \times D_{\text{ff}} = 4 \times B \times S \times H \times D_{\text{ff}}$$

- **Output projection:**

$$\text{FLOPs}_{\text{out}} = \text{FLOPs}_{\text{out}_{\text{proj}}} + \text{FLOPs}_{\text{softmax}} = 2 \times B \times S \times H \times V + 3 \times B \times S \times V$$

**Total Forward FLOPs per Layer:**

$$\text{FLOPs}_{\text{layer}} = (\text{FLOPs}_{\text{MHA}} + \text{FLOPs}_{\text{FFN}}) \times L$$

**Total Forward FLOPs per Step:**

$$\text{FLOPs}_{\text{forward}} = \text{FLOPs}_{\text{emb}} + \text{FLOPs}_{\text{layer}} + \text{FLOPs}_{\text{out}}$$

### C.2.2    BACKWARD

As discussed in Appendix C.1, for the backward computation all slices contribute to the gradients with respect to the input.

**MHA backward**    Let $\rho_{\text{attn}} = \frac{1}{N}$ be the trained fraction of heads. Using the forward costs:

$$\text{FLOPs}_{\text{proj}} = 6 \times B \times S \times H^2, \quad \text{FLOPs}_{\text{attn}} = 4 \times B \times S^2 \times H, \quad \text{FLOPs}_{\text{out\_proj}} = 2 \times B \times S \times H^2,$$

the backward splits as follows:

$$\begin{aligned}
\text{(i) Output projection } \mathbf{W}_O: \quad & \text{input Jacobian: } 2 \times B \times S \times H^2, \\
& \text{parameter gradient: } 2 \times B \times S \times H^2, \\
\text{(ii) Attention matmuls } (\mathbf{QK}^\top \text{ and } \mathbf{AV}): \quad & \text{backward } \approx 2 \times \text{FLOPs}_{\text{attn}} = 8 \times B \times S^2 \times H, \\
\text{(iii) Q/K/V projections } (\mathbf{W}_Q, \mathbf{W}_K, \mathbf{W}_V): \quad & \text{input Jacobian: } 2 \times \text{FLOPs}_{\text{proj}}^{(\text{half})} = 6 \times B \times S \times H^2, \\
& \text{parameter gradients: } 2 \times \text{FLOPs}_{\text{proj}}^{(\text{half})} \times \rho_{\text{attn}}.
\end{aligned}$$

$$\text{FLOPs}_{\text{MHA}}^{\text{bwd}} = \underbrace{8 \times B \times S^2 \times H}_{\text{attn matmuls}} + \underbrace{(10 + 6 \times \rho_{\text{attn}}) \times B \times S \times H^2}_{\substack{\text{Q/K/V input Jacobian (6)} \\ \text{+ Q/K/V param grads } (6\rho_{\text{attn}}) \\ \text{+ } \mathbf{W}_O \text{ input Jacobian (2)} \\ \text{+ } \mathbf{W}_O \text{ param grad (2)}}}.$$

When $\rho_{\text{attn}} = 1$ (no freezing), this reduces to the usual "backward $\approx 2\times$ forward" for MHA:

$$\text{FLOPs}_{\text{MHA}}^{\text{bwd}}(\rho_{\text{attn}}{=}1) = 8 \times B \times S^2 \times H + 16 \times B \times S \times H^2 = 2 \times (4 \times B \times S^2 \times H + 8 \times B \times S \times H^2).$$

**FFN backward**    Similarly as for MHA for FFN let $\rho_{\text{mlp}} = \frac{1}{N}$. Then:

$$\text{FLOPs}_{\text{FFN}}^{\text{bwd}} = \underbrace{4 \times B \times S \times H \times D_{\text{ff}}}_{\text{full input Jacobian}} + \underbrace{4 \times \rho_{\text{mlp}} \times B \times S \times H \times D_{\text{ff}}}_{\text{parameter gradients}}$$

This results in total backward per step:

$$\text{FLOPs}_{\text{step}}^{\text{bwd}} = 2 \times \text{FLOPs}_{\text{emb}} + L \times (\text{FLOPs}_{\text{MHA}}^{\text{bwd}} + \text{FLOPs}_{\text{FFN}}^{\text{bwd}}) + 2 \times \text{FLOPs}_{\text{out}}.$$

# D  ADDITIONAL RESULTS

As discussed in Appendix C.1, the only gradients we omit are those with respect to frozen parameters. While this reduces FLOPs during the backward pass, it does not decrease activation memory. A natural question is whether one could go further — not only skipping parameter gradients but also fully detaching their contributions from the gradient flow. We therefore investigated how such a complete detachment of frozen MLP slices affects convergence.

**Backward detach-all-but-$k$**   In this setting, the forward pass still uses all parameters, but during backpropagation we compute only the Jacobian components corresponding to the active slice.

Backward with zeroed slices (only slice $k$ active):

$$\widetilde{\mathbf{A}}_n = \begin{cases} \mathbf{A}_k, & n = k, \\ \mathbf{0}, & n \neq k, \end{cases}$$

$$\widetilde{\mathbf{M}}_n = \begin{cases} \mathbb{I}(\mathbf{H}_k > 0), & n = k, \\ \mathbf{0}, & n \neq k, \end{cases}$$

where $\mathbf{M}_n = \mathbb{I}(\mathbf{H}_n > 0)$ is the elementwise activation mask (e.g., ReLU).

Given upstream $\mathbf{G} = \frac{\partial \mathcal{L}}{\partial \mathbf{Y}} \in \mathbb{R}^{d \times m}$:

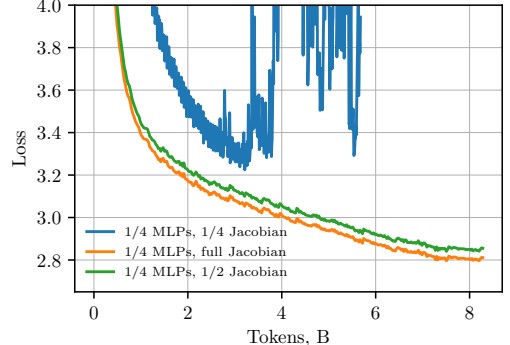

Figure 6: Training loss as a function of training tokens for a 335M-parameter model trained with the 1/4 MLP slicing strategy described in § 2.2.1. *Full Jacobian* denotes our standard approach where all slices contribute to the Jacobian. *1/4 Jacobian* corresponds to the detach-all-but-$k$ variant, where only the $k$-th slice contributes. *1/2 Jacobian* refers to the $k$+random variant, where the $k$-th slice and one additional randomly selected slice are retained.

$$\frac{\partial \mathcal{L}}{\partial \mathbf{V}_n} = \mathbf{G}\,\widetilde{\mathbf{A}}_n^\top = \begin{cases} \mathbf{G}\,\mathbf{A}_k^\top, & n = k, \\ \mathbf{0}, & n \neq k, \end{cases}$$

$$\frac{\partial \mathcal{L}}{\partial \mathbf{A}_n} = \mathbf{V}_n^\top \mathbf{G}, \qquad \frac{\partial \mathcal{L}}{\partial \mathbf{H}_n} = \left( \frac{\partial \mathcal{L}}{\partial \mathbf{A}_n} \right) \odot \widetilde{\mathbf{M}}_n = \begin{cases} (\mathbf{V}_k^\top \mathbf{G}) \odot \mathbb{I}(\mathbf{H}_k > 0), & n = k, \\ \mathbf{0}, & n \neq k, \end{cases}$$

The full-Jacobian input gradient is

$$\frac{\partial \mathcal{L}}{\partial \mathbf{X}} = \sum_{n=1}^N \mathbf{W}_n^\top \left( \frac{\partial \mathcal{L}}{\partial \mathbf{H}_n} \right) = \sum_{n=1}^N \mathbf{W}_n^\top \left( (\mathbf{V}_n^\top \mathbf{G}) \odot \mathbf{M}_n \right). \tag{6}$$

Under the detach-all-but-$k$ rule, this can be written as single-slice contribution:

$$\boxed{\frac{\partial \mathcal{L}}{\partial \mathbf{X}} = \mathbf{W}_k^\top \left( (\mathbf{V}_k^\top \mathbf{G}) \odot \mathbb{I}(\mathbf{H}_k > 0) \right)}$$

since $\frac{\partial \mathcal{L}}{\partial \mathbf{H}_n} = \mathbf{0}$ for all $n \neq k$.

As expected, since the full backward pass is disrupted, naively detaching all but one slice resulted in gradient explosion in the middle of training (Figure 6).

**Backward detach-all-but-$k$ + random**   To study this further, we considered a variant where, instead of keeping only the $k$-th slice, we retain the $k$-th slice *plus* one additional slice chosen at random. Concretely, during the forward pass we randomly sample an index $g \sim \text{Unif}(\{1, \ldots, N\} \setminus \{k\})$ and keep the corresponding contributions in the backward Jacobian.

Given the full Jacobian in Eq. 6,

$$\frac{\partial \mathcal{L}}{\partial \mathbf{X}} = \sum_{n=1}^N \mathbf{W}_n^\top \left( (\mathbf{V}_n^\top \mathbf{G}) \odot \mathbb{I}(\mathbf{H}_n > 0) \right),$$

the detach-all-but-$k$+random variant reduces this to

$$\frac{\partial \mathcal{L}}{\partial \mathbf{X}} = \mathbf{W}_k^\top \left( (\mathbf{V}_k^\top \mathbf{G}) \odot \mathbb{I}(\mathbf{H}_k > 0) \right)$$
$$+ \mathbf{W}_g^\top \left( (\mathbf{V}_g^\top \mathbf{G}) \odot \mathbb{I}(\mathbf{H}_g > 0) \right),$$

where $g$ is resampled independently at each step.

Despite this modification, performance still degraded (perplexity 17.38 vs. 16.44). Further investigation of how much of the Jacobian can be dropped could be an interesting direction for future work. Adjusting hyperparameters, or scaling the activations might improve the performance. Also better strategies than picking uniformly at random can be explored.

