# OpenReview forum: "Partial Parameter Updates for Efficient Distributed Training"
_ICLR.cc/2026/Conference — Submitted to ICLR 2026_

### Official Review · Reviewer_twfF · 2025-10-27

**Soundness:** 3
**Presentation:** 3
**Contribution:** 2
**Rating:** 6
**Confidence:** 3

**Summary:**

This paper presents a partial parameter updates scheme based on the DiLoCo framework, which reduces memory use and communication costs by freezing parameters on each node. Unlike Streaming DiLoCo which divide the model in a pipeline-parallel manner, this work slice the training in a tensor-parallel way to achieve a similar results.

**Strengths:**

This paper is well-structured and the idea is easy to follow, and it provides targeted optimizations to address the problem of high peak memory consumption in Streaming DiLoCo. Under low-bandwidth settings, proposed method converges faster than standard DDP.

**Weaknesses:**

1. It looks like a natural extension to Streaming DiLoCo with frozen weights, also lacks convergence guarantee in theory. As seen in Table 1, the ppl goes up quickly when N is larger than 8.
2. The improvement compared to Streaming DiLoCo is marginal, seen in Figure 3, while the curve for Streaming DiLoCo is smoother, indicating a more stable run.

**Questions:**

1. Could you provide experimental results for a Streaming DiLoCo synchronization strategy that has the same communication budget as 1/4 MLP? I think it might be a better baseline when comparing convergence.
2. Is there any clear pattern in the strategy selection for parameter slicing (eg. training a model with different shape), and could you provide ablation experiments that only slice attention heads?

---

> ### Author Response · Authors · 2025-11-27
> **Response to Reviewer twfF**
>
> We thank the reviewer for acknowledging the strengths of our work and for raising questions that helped us improve the paper.
>
> >It looks like a natural extension to Streaming DiLoCo with frozen weights
>
> We agree that our method can be seen as a natural extension of Streaming DiLoCo with frozen weights. However, we view this simplicity as a feature: it makes the method easy to implement and adapt to different settings. At the same time, as shown in Table 1, it provides substantial memory and compute savings with no observed degradation in performance.
>
> > ...also lacks convergence guarantee in theory
>
> We would like to note that DiLoCo and Streaming DiLoCo also do not provide theoretical convergence guarantees, so our method is comparable to prior work in this respect.
>
> >As seen in Table 1, the ppl goes up quickly when N is larger than 8.
>
> It is true that perplexity increases, but this reflects a fundamental trade-off between convergence speed and memory savings. When we slice into 8 partitions, each parameter receives 8 times fewer gradient updates per step, so more training steps are needed to converge. In Table 1, we report an additional experiment with 1/8 MLP slicing trained on 37B tokens, which shows that with more training our 1/8 configuration matches the performance of the 1/4 MLP configuration. This aggressive slicing enables training larger models on devices with limited RAM without offloading the optimizer state at every step, this is a fundamental trade-off that we intentionally make.
>
> >The improvement compared to Streaming DiLoCo is marginal, seen in Figure 3, while the curve for Streaming DiLoCo is smoother, indicating a more stable run.
>
> We agree that the FLOP improvement is modest. However, even marginal FLOP savings at fixed final performance imply that our method converges faster in terms of wall-clock time. In addition, our method yields significant memory savings (Figure 2(a)), which is one of our primary goals.
> Regarding “stability”, we find it difficult to draw strong conclusions based solely on the visual smoothness of the training curves, as there is no clear, quantitative notion of stability in this context. In our experiments, we did not observe systematic instabilities such as divergence or training collapse for our method compared to Streaming DiLoCo.
>
> >Could you provide experimental results for a Streaming DiLoCo synchronization strategy that has the same communication budget as 1/4 MLP?
>
> The communication budget of our method and Streaming DiLoCo / DiLoCo is identical (as stated in Abstract P.1, L.021). Although the delta in our method is sparse, the communication budget remains the same due to how ring all-reduce is implemented. We stated it clearly in revised version (Section 3.2, Convergence Speed Under Bandwidth Constraints, P. 7 ).
>
> > Is there any clear pattern in the strategy selection for parameter slicing (e.g., training a model with different shape), and could you provide ablation experiments that only slice attention heads?
>
> As described in Section 2.2.1, P.5, L239-242 and Appendix A.2, the strategy we chose is primarily driven by the simplicity of implementation for the forward and backward passes and the fact that MLP layers contain the majority of the model parameters. Therefore, it is natural from our standpoint to slice MLPs so that we can save more memory and FLOPs.
> Regarding slicing only attention heads, we thank the reviewer for expressing interest. Here are the corresponding results, which we added to Table 1 of the paper and clarified the result in Section 3.2.
> | Method    | Perplexity | Memory (GB) | Trainable parameters (B) | Tokens (B) |
> |----------:|-----------:|------------:|--------------------------:|-----------:|
> | 1/4 heads |     12.83  |      16.95  |                     10.7  |        26  |
>
> Although the performance is similar to 1/4 MLP slicing, the number of active parameters (and therefore the memory usage) is higher.

---

### Official Review · Reviewer_CRF3 · 2025-10-29

**Soundness:** 2
**Presentation:** 3
**Contribution:** 2
**Rating:** 2
**Confidence:** 3

**Summary:**

The paper introduces a method to improve the efficiency of distributed Transformer training. Specifically, it freezes a subset of parameters on each node, allowing local updates and an all-reduce operation on the local weight changes (similar to DiLoCo), followed by an outer optimization step. By omitting optimizer states for the frozen parameters, the approach reduces memory usage and achieves better performance per FLOP.

**Strengths:**

The proposed method in the paper is both interesting and practical. Since adaptive optimizer states typically consume more memory than the model parameters themselves, the resulting memory savings are substantial. These savings can be leveraged to train larger models or use bigger batch sizes, leading to more efficient resource utilization.

The paper is well written, and the proposed approach is simple yet effective. It can be easily implemented with existing communication-efficient distributed training frameworks. The experimental results are promising and demonstrate the potential of the method in improving both memory efficiency and training performance.

**Weaknesses:**

While the proposed method is promising, its main limitation lies in the lack of experimental breadth. Evaluating only a single model size, in a single distributed configuration, and on a single dataset is insufficient to convincingly demonstrate the method’s generality or robustness. Additional ablations and experiments under diverse settings would significantly strengthen the paper’s claims.

Moreover, the reported memory savings are highly dependent on the choice of optimizer. For instance, modular optimizers such as Muon [1] and its variants are gaining traction and have shown that momentum-based SGD can perform comparably to adaptive optimizers like AdamW. In such cases, the memory advantage offered by the proposed approach becomes less substantial.

[1] https://arxiv.org/abs/2502.16982

**Questions:**

1. It would be valuable to include an ablation showing how the proposed method performs as the number of nodes (or model replicas) increases. This would clarify how communication efficiency and performance scale with distributed configurations.

2. Including training and validation curves plotted against the number of processed tokens for all compared baselines would be helpful. Such curves would help assess convergence behavior, stability, and sample efficiency of the method.

3. Including the performance of standard DDP training in Table 1 would provide a stronger reference point and make it easier to contextualize the improvements of the proposed approach.

4. Additional experiments across multiple datasets, network depths, and hidden dimension sizes would significantly strengthen the empirical section. These would help demonstrate that the observed gains are consistent across varying model scales and data regimes.

---

> ### Author Response · Authors · 2025-12-03
>
> We thank the reviewer for finding our method interesting and practical, and for acknowledging that our experimental results are promising and demonstrate the potential of the method to improve both memory efficiency and training performance.
>
> > Additional ablations and experiments under diverse settings would significantly strengthen the paper’s claims.
>
> Thank you for pointing this out. In the revised version, we add results for a 2.6B-parameter model trained on 64 GPUs (doubling the original configuration) on 52B tokens. We believe this larger-scale experiment strengthens the empirical support for our method and its applicability beyond 1.3B/32-node setting.
>
> > Moreover, the reported memory savings are highly dependent on the choice of optimizer. For instance, modular optimizers such as Muon [1] and its variants are gaining traction and have shown that momentum-based SGD can perform comparably to adaptive optimizers like AdamW. In such cases, the memory advantage offered by the proposed approach becomes less substantial.
> [1] https://arxiv.org/abs/2502.16982
>
> We fully agree with this observation and already discuss efficient optimizers in the related work section (p. 9, l. 474). Using optimizers with smaller state (e.g., momentum-only methods) would indeed reduce the absolute memory savings of our approach. However, this is largely orthogonal to the main contribution of our method: our slicing scheme reduces the amount of trainable parameters per node and thus the optimizer state and compute, independently of the specific optimizer used.
>
> > It would be valuable to include an ablation showing how the proposed method performs as the number of nodes (or model replicas) increases. This would clarify how communication efficiency and performance scale with distributed configurations.
>
> In the revised version, we include an additional experiment on 64 compute nodes. We hope this larger configuration helps illustrate how our method behaves as the distributed setup scales, and addresses the reviewer’s concern about the dependence on the number of nodes.
>
> > Including training and validation curves plotted against the number of processed tokens for all compared baselines would be helpful. Such curves would help assess convergence behavior, stability, and sample efficiency of the method.
>
> In our setup, the number of processed tokens is identical for Streaming DiLoCo (baseline) and for our method (as specified in the experiment setup), and we already report final performance at the same token budget in Table 1. In our experiments, the loss–vs–tokens curves for the two methods are very similar, so including them would not change the main message of the paper. Due to space constraints, we chose instead to report loss–vs–FLOPs, which better highlights the efficiency gains we aim to demonstrate.
>
> > Including the performance of standard DDP training in Table 1 would provide a stronger reference point and make it easier to contextualize the improvements of the proposed approach.
>
> Thank you for the suggestion. We will add this line in the revised paper.
>
> > Additional experiments across multiple datasets, network depths, and hidden dimension sizes would significantly strengthen the empirical section. These would help demonstrate that the observed gains are consistent across varying model scales and data regimes.
>
> We agree that more experiment results would further strengthen the paper. In the revised version, we partially address this by adding results for a larger 2.6B-parameter model (with increased depth and hidden size) trained on 64 GPUs, in addition to the original 1.3B/32-GPU configuration (Table 1). Both configurations are trained on RedPajama-V2, a large and diverse multilingual web corpus, which we believe is a representative setting for large-scale pre-training. For both configurations, we observe memory and compute savings without any loss in performance compared to prior work. Extending the study to additional datasets and architectures is a natural direction for future work.

---

### Official Review · Reviewer_H2UE · 2025-10-31

**Soundness:** 3
**Presentation:** 3
**Contribution:** 1
**Rating:** 4
**Confidence:** 4

**Summary:**

This paper introduces an efficient algorithms for low-communication distributed data-parallel training that performs local updates on a node-specific subset of parameters. The author(s) validate the effectiveness of the method by training a 1.3B LLMs on 32 nodes and compare it with Streaming DiLoCo on Perplexity and training FLOPs.

**Strengths:**

1. This paper describes the research method in detail.
2. This paper describes the experimental setup in detail.

**Weaknesses:**

1. There is no discussion on block coordinate optimization. There is plenty of literacture on block coordinate optimization [1-3] and it is unclear what is the main technical contribution of the paper.
2. There is no comparison with block coordinate optimization and parameter-efficient fine-tuning (PEFT) in the experiments. Given that there aer a lots of existing methods on block coordinate optimization, they should be included as baselines. Meanwhile, the proposed method has similar performance advantages to PEFT methods such as LoRA [4], requiring only communicating matrices smaller than the model weights during parameter synchronization. So PEFT methods should be included in the comparison.

[1] Accelerating Block Coordinate Descent for LLM Finetuning via Landscape Correction

[2] Memory-Efficient Block Coordinate Descent for Hessian-Informed Zeroth-Order Optimizer

[3] How to Train a Model on a Cheap Cluster with Low Cost using Block Coordinate Descent

[4] Lora: Low-rank adaptation of large language models.

**Questions:**

N/A

---

> ### Author Response · Authors · 2025-11-27
> **Response to Reviewer H2UE**
>
> We would like to thank the reviewer for reading our paper and for providing additional references.
>
> > There is plenty of literacture on block coordinate optimization [1-3] and it is unclear what is the main technical contribution of the paper. There is no comparison with block coordinate optimization
>
> [1] As stated on page 14, line 707 of [1]:
>
> *We use DeepSpeed ZeRO-3 to implement all the experiments that require distributed training.*
>
> The authors use ZeRO-3/FSDP-style sharding, which **communicates at every step and assumes a high-bandwidth interconnect**. There is no discussion of low-communication or bandwidth-limited scenarios. Since our objective is precisely to study training under limited bandwidth (Introduction, line 78), the setting of [1] is different from ours, and we do not view it as a directly comparable baseline.
>
> [2] This paper proposes a new block coordinate optimization method for fine-tuning (line 102) on a single consumer GPU (line 91). There is **no multi-node communication and no low-bandwidth constraint**, the focus is on adapting an already trained model on one device. This is a very different setting from the low-communication, multi-node pre-training scenario we target.
>
> [3] We thank the reviewer for this reference and have added it to the related work section. The authors employ pipeline parallelism, which differs from our setup, as our approach is purely data-parallel. In addition, this study does not report explicit results in a low-bandwidth regime. As we note in the related work (P. 9, L.460 - 462), our approach could in principle be extended to pipeline-parallel settings. We therefore view [3] as complementary to our method rather than as a directly comparable baseline, and a full empirical comparison is beyond the scope of this work.
>
> > Meanwhile, the proposed method has similar performance advantages to PEFT methods such as LoRA [4], requiring only communicating matrices smaller than the model weights during parameter synchronization. So PEFT methods should be included in the comparison.
>
> Can the reviewer please clarify in more detail the similarities they see between our method and PEFT approaches?
> As we state throughout the paper (Abstract, L.12–16; Introduction, L.88–91), our goal is not fine-tuning but full-parameter distributed pre-training in a low-communication setting. This is fundamentally different from the typical PEFT/adapter setting: in our regime, all weights must eventually receive updates, and the question we explore is whether this can be done more efficiently than prior low-communication data-parallel approaches. For this reason, we do not consider adapter-based fine-tuning methods to be directly relevant baselines, although our method could, in principle, be extended to the fine-tuning case.
>
> [4] Similarly to [1] and [2], [4] (LoRA) is a PEFT method which main goal is to make fine-tuning feasible on a consumer GPU by updating a small number of additional parameters while keeping the backbone frozen. Therefore [4] is a very different setting from the low-communication, multi-node pre-training scenario we target.

---

### Official Review · Reviewer_rLg8 · 2025-11-04

**Soundness:** 3
**Presentation:** 3
**Contribution:** 2
**Rating:** 4
**Confidence:** 4

**Summary:**

This paper proposes partial parameter updates for local SGD and then global synchronization with diloco/fedavg to reduce communication cost between workers in a data parallel setting. The core idea is that each worker optimises a subset of model parameters (predetermined) and other frozen parameters are only updated through the outer optimizer after the global sync. The experiments show comparable performance with streaming diloco wrt to flops.

**Strengths:**

1. Even though incremental, the idea is simple and seems effective. The way of determining the subsets is similar to MoE and makes sense. Due to partial parameter updates, per-node memory usage is reduced, enabling larger models to be trained in the DP setting.
2. Language model experiment shows performance matching streaming diloco when compared against flops processed.

**Weaknesses:**

1. Limited experiments: Only one dataset and one datasets and one baseline method compared. Specifically, sparta (Beton et al) is a relevant baseline, and diloco has many variants -- for instance, outer gradients can be compressed using quantization/low-rank to reduce communication cost. These methods could be compared to correctly position the paper wrt convergence speed and communication cost.
    - As the main idea is heuristic and experiments with multiple datasets and model sizes would strengthen the paper.
2. Even though wrt flops, the results match streaming diloco, wrt to training steps it is slow.
3. To reduce memory usage and fit larger models into smaller gpus, one can adopt pipeline parallelism with low-rank compression of activations and gradients as shown in [1]. This is a relevant comparison on the aspect of fitting larger models in smaller gpus.

[1] Ramasinghe, Sameera, et al. "Protocol Models: Scaling Decentralized Training with Communication-Efficient Model Parallelism." arXiv preprint arXiv:2506.01260 (2025).

**Questions:**

1. Fig.1 shows that the outer optimizer only updates the frozen parameters, but this contradicts line 15 of Algorithm 1. Also, there are common trainable parameters between workers, so they will also be updated during the outer step. Please clarify

---

> ### Author Response · Authors · 2025-11-27
> **Response to rLg8**
>
> We thank the reviewer for reading our paper, pointing out its weaknesses, and asking questions that helped us improve it.
>
> > Limited experiments: Only one dataset and one datasets and one baseline method compared.
>
> Thank you for pointing this out. We agree that additional experiments strengthen the paper. In the revised version, Table 1 includes results for 1/4 MLP slicing on a 2.6B model trained on 64 GPUs on the RedPajama-V2 dataset.
>
> We would like to note that the original DiLoCo paper reports results on a single dataset (C4). RedPajama-V2 is a large, diverse, multilingual web corpus, so we believe that pretraining on this dataset is sufficient to demonstrate that our approach works in a realistic large-scale setting. If the reviewer has specific additional datasets in mind that they consider important for evaluation, we would greatly appreciate the suggestion.
>
> > Specifically, Sparta (Beton et al.) is a relevant baseline.
>
> We respectfully disagree that Sparta is a directly relevant baseline for our setting and we explained this in the original submission in the Related Work section (P. 9). The method proposed by Beton et al. (2025) reduces divergence across nodes, however, all parameters are still updated on every device, meaning each node must store the full optimizer state and perform full backpropagation. In contrast, our method updates only a fixed subset of parameters per node, which directly reduces both memory and compute.
>
> >…and DiLoCo has many variants — for instance, outer gradients can be compressed using quantization/low-rank to reduce communication cost.
>
> We fully agree that quantization or low-rank compression of outer gradients is a natural way to reduce communication cost. However, the goal of our paper is not to improve the communication budget: by design, our communication cost is identical to DiLoCo and Streaming DiLoCo. As we state throughout the paper, our aim is to make low-communication methods with infrequent synchronization more memory- and compute-efficient. Moreover, in Streaming DiLoCo and DiLoCo, the outer gradients are accumulated in full precision locally, so quantizing the communicated outer gradients would not change the memory usage or local compute. For this reason, adding a quantized-outer-gradient variant would not directly address the main question of our paper.
>
> > Even though wrt FLOPs, the results match Streaming DiLoCo, wrt training steps it is slow.
>
> We are not entirely sure which notion of “slow” the reviewer refers to here. In our experiments, Streaming DiLoCo and our method are trained for the same number of steps, with the same batch size and the same total number of tokens, so the number of steps required to reach a given token budget is identical by construction (Table 1; Experiment setup, p. 7). Perplexity as a function of tokens/steps is also closely matched between the two methods (Table 1).
>
> > To reduce memory usage and fit larger models into smaller GPUs, one can adopt pipeline parallelism with low-rank compression of activations and gradients as shown in [1]. This is a relevant comparison on the aspect of fitting larger models in smaller GPUs.
> [1] Ramasinghe, Sameera, et al. "Protocol Models: Scaling Decentralized Training with Communication-Efficient Model Parallelism." arXiv preprint arXiv:2506.01260 (2025).
>
> We thank the reviewer for pointing us to [1], we have added it to the related work section. We agree that pipeline parallelism with activation compression is a powerful way to fit larger models on smaller GPUs. However, as we mention in the related work, pipeline parallelism is orthogonal to our setting: we focus on data-parallel training which is a different parallelisation axis. Therefore we believe that  implementing and tuning method proposed in [1]  is beyond the scope of this paper. We made it more explicit in the revised version (P.9, L.464)
>
> > Fig.1 shows that the outer optimizer only updates the frozen parameters, but this contradicts line 15 of Algorithm 1. Also, there are common trainable parameters between workers, so they will also be updated during the outer step. Please clarify
>
> Thank you for catching this. You are correct that, in Algorithm 1, the outer optimizer updates all parameters, including those that are locally trainable on some workers, because the same slice can be trained on multiple nodes and their updates are aggregated. In Fig. 1, we show a simplified example where the number of slices equals the number of devices and each half of the parameters is trained only on one device, in that particular illustration, the accumulated outer update on a given device appears only on parameters that were frozen locally. We realized that it can be misleading. In the revised version, we updated Fig. 1.

---

### Author Response · Authors · 2025-12-03
**Message to AC: Rebuttal Summary**

We would like to thank the AC and all reviewers for their time. Unfortunately, there was no rebuttal discussion. Below is a brief summary of the rebuttal:

### **Reviewer rLg8**
- We addressed the “limited experiments” concern by adding a larger-scale setting (1.3B -> 2.6B, 32 -> 64 GPUs, 26B -> 52B tokens; Table 1).
 - We clarified that suggested baselines (quantization, pipeline parallelism) are orthogonal to our method and were already discussed in the paper (Related work, p. 9).
  - The claim reviewer made that our method is “slow wrt training steps” is incorrect.
### **Reviewer H2UE**
- The reviewer asks for comparisons to methods for parameter-efficient fine-tuning **on a single GPU / high-bandwidth setting**, which is very different regime from the **low-communication, multi-node pre-training scenario** we target. Therefore, we disagree that these baselines are relevant.
### **Reviewer CRF3**
- We were very surprised by the low score, given the very positive tone of the review and the absence of any clearly discouraging remarks. The reviewer mentioned:

    >The proposed method in the paper is both interesting and practical...
   > The paper is well written, and the proposed approach is simple yet effective. The experimental results are promising and demonstrate the potential of the method in improving both memory efficiency and training performance.

- The main concern was limited experiments. We believe the added 2.6B / 64-GPU experiment directly addresses this.
- The comment about optimiser is already discussed in related work (p. 9, line 475): our method is orthogonal to the optimizer choice.
### **Reviewer twfF**
   - We added the requested ablation with head-only slicing (Table 1).
   - We clarified that the communication budget of our method and Streaming DiLoCo / DiLoCo is identical.
   - We noted that, despite modest FLOPs savings, even ~15% implies faster convergence in wall-clock time.

---

### Meta-Review · Area_Chair_boDq · 2026-01-10

**Summary:**

This paper proposes a simple extension to DiLoCo-style low-communication training by performing local updates on only a subset of parameters, with the goal of reducing peak memory usage and training FLOPs.

Reviewers generally found the idea simple and practical, but all reviewers raised concerns about the limited empirical validation of the proposed method. While additional experiments were added during the rebuttal phase, the current version still does not convincingly establish (a) the practical benefit of the proposed approach relative to existing communication-efficient training techniques, (b) how the improvement scales with distributed configurations such as the number of nodes, or (c) whether the method consistently improves performance across different datasets and architectures.

**Reviewer Concerns:**

* Reviewer rLg8 suggested several important baselines, including variants of DiLoCo with quantization or low-rank compression, as well as pipeline parallelism with low-rank compression. While the authors argue that these techniques are orthogonal to their approach, these variants may already substantially reduce communication cost, and the rebuttal does not resolve how much additional benefit partial parameter updates would provide when such methods are applied.

* Reviewer H2UE suggested comparisons with block coordinate optimization and parameter-efficient fine-tuning methods, but did not identify specific baselines targeting communication-efficient pretraining. The authors' rebuttal clarifies that these suggested comparisons are outside the scope of the paper, and thus this concern is largely addressed.

* Reviewer CRF3 requested a more systematic study of the scaling behavior as the number of nodes increases. Although the authors added an additional experiment with 64 GPUs during the rebuttal phase, the evaluation remains insufficient to analyze scaling trends. Reviewer CRF3 also suggested experiments across multiple datasets and architectures to assess the robustness of the results, which were not added in the revision.

* Reviewer twfF noted that the improvement in loss over Streaming DiLoCo appears marginal. This concern was almost addressed in the rebuttal, where the authors argue that the observed improvement is enough to imply lower wall-clock time. The reviewer also asked for results using a Streaming DiLoCo synchronization strategy with the same communication budget as 1/4 MLP slicing, and this was addressed by the authors' clarification that the communication cost is already identical. Finally, the authors provided an ablation with head-only parameter slicing, as requested by Reviewer twfF. The remaining concern raised by Reviewer twfF is the lack of theoretical convergence guarantees and the empirical observation that perplexity increases quickly as the model is partitioned into more slices.

**Reviewer Scores:**

* Reviewer rLg8 is likely to keep the score at 4, as the main concern regarding limited comparison with stronger baselines remains unresolved.

* Reviewer H2UE may increase the score from 4 to 6, as the rebuttal largely addresses the reviewer's concerns.

* Reviewer CRF3 is likely to keep the score at 2, since the main concerns regarding empirical breadth and robustness are not fully addressed.

* Reviewer twfF is likely to keep the score at 6, as their concerns are only partially addressed.

---

### Decision · Program_Chairs · 2026-01-26

Reject